# Vibrotactile Feedback for a Person with Transradial Amputation and Visual Loss: A Case Report

**DOI:** 10.3390/medicina59101710

**Published:** 2023-09-25

**Authors:** Gerfried Peternell, Harald Penasso, Henriette Luttenberger, Hildegard Ronacher, Roman Schlintner, Kara Ashcraft, Alexander Gardetto, Jennifer Ernst, Ursula Kropiunig

**Affiliations:** 1Rehabilitation Clinic Tobelbad, Austrian Workers’ Compensation Board (AUVA), 8144 Tobelbad, Austria; 2Ludwig Boltzmann Institute for Traumatology, 1200 Vienna, Austria; 3Saphenus Medical Technology GmbH, 2500 Baden, Austria; 4Department of Integrative Physiology, University of Colorado Boulder, Boulder, CO 80309, USA; kara.ashcraft@colorado.edu; 5Department of Plastic, Aesthetic and Reconstructive Surgery with Hand Surgery and Competence Center for Bionic Prosthetics, Brixsana Private Clinic, 39042 Bressanone, Italy; alexander.gardetto@brixsana.it; 6Department of Trauma Surgery, Hannover Medical School, 30625 Hanover, Germany; ernst.jennifer@mh-hannover.de

**Keywords:** peripheral phantom limb map, somatosensory feedback, visual control, blindness, vibration

## Abstract

*Background and Objectives:* After major upper-limb amputation, people face challenges due to losing tactile information and gripping function in their hands. While vision can confirm the success of an action, relying on it diverts attention from other sensations and tasks. This case report presents a 30-year-old man with traumatic, complete vision loss and transradial left forearm amputation. It emphasizes the importance of restoring tactile abilities when visual compensation is impossible. *Materials and Methods:* A prototype tactile feedback add-on system was developed, consisting of a sensor glove and upper arm cuff with related vibration actuators. *Results:* We found a 66% improvement in the Box and Blocks test and an overall functional score increase from 30% to 43% in the Southampton Hand Assessment Procedure with feedback. Qualitative improvements in bimanual activities, ergonomics, and reduced reliance on the unaffected hand were observed. Incorporating the tactile feedback system improved the precision of grasping and the utility of the myoelectric hand prosthesis, freeing the unaffected hand for other tasks. *Conclusions:* This case demonstrated improvements in prosthetic hand utility achieved by restoring peripheral sensitivity while excluding the possibility of visual compensation. Restoring tactile information from the hand and fingers could benefit individuals with impaired vision and somatosensation, improving acceptance, embodiment, social integration, and pain management.

## 1. Introduction

Losing a hand or arm can significantly affect a person’s life. It is estimated that, as of 2020, there are more than 2 million people living with limb amputation in the United States, and of those 2 million, approximately 700,000 people have experienced upper-limb loss [1]. Transradial amputations account for 40% of major above-wrist upper-limb amputations [2], with 70% of these amputations being trauma-related, from accidents involving machinery or explosions, and requiring the use of a prosthetic limb [2,3]. Significant advances have been made in developing new prosthetic technologies, with the goal of restoring function to people with transradial amputation. Despite these technological advances, many challenges remain in understanding obstacles in patient adoption of technology, and what critical factors should be focused on in prosthetic development in order to return function and sensitivity to the amputated limb.

Tasks that require precise finger movements, like buttoning a shirt, or more complex movements, like holding a cup, can pose difficulties for people who experience loss of peripheral sensitivity due to upper-limb amputation. Further, explosion events, particularly in the context of war, bombs, and firecracker use, often lead to concomitant injuries such as visual impairment or blindness, increasing the reliance on peripheral sensitivity to perform tasks of daily living [4,5,6]. In the United States, visual loss affects around 0.3% of the population [7]. Of 18 to 64 year olds in the US population, 0.6% reported significant vision loss [8], of which Carty et al. [9] estimated that 0.3% potentially have concomitant limb amputations. Peripheral sensitivity plays a crucial role in sensorimotor control, contributing to almost 50% of the overall function of the hand [10], and thus has important applications in the design and use of prostheses.

Without a dedicated sensory feedback system, people with upper-limb amputation rely on indirect somatosensory information perceived through tactile cues, such as detecting the bending and resistance of the prosthesis socket while manipulating objects. To our knowledge, no commercially available somatosensory feedback system for upper-limb prostheses is on the market [11,12,13] and restoring the sensitivity of the hand remains a challenge [14,15,16,17]. Although several myoelectric multi-grip hands are currently available [18], these devices lack tactile feedback from the affected hand. Sensory feedback in upper-limb prostheses was shown to enhance motor control by providing vital information about the limb’s position, movement, and contact forces. This led to more natural movements, better functionality during everyday tasks, and less cognitive strain for the user [19]. Invasive solutions have been explored, such as direct nerve stimulation using intraneural electrodes [16] and the use of implanted technologies for neural interfaces in surgically prepared nerve–muscle grafts [16,20,21,22,23], which come with surgical risks of nerve damage and challenges in long-term reliability [24,25]. Previous non-invasive vibrotactile feedback approaches were designed to translate grip force into vibration cues embedded in the prosthesis. However, this approach does not provide spatial information, which is needed to improve grasping, holding, and releasing objects, while also reducing reliance on visual control [11,26,27]. In addition to improved prosthetic dexterity, somatosensory feedback for people with upper-limb amputation has been shown to improve the utility and acceptance of the prosthesis, thus increasing social participation and quality of life [11,26,28], as well as treating the perception of pain in amputated areas of the limb, known as phantom limb pain [26,29,30,31,32].

A promising development in the restoration of peripheral sensitivity in people with transradial amputation is a phenomenon called a peripheral phantom limb map [33,34,35]. Phantom limb maps are a phenomenon experienced by some persons with amputation, where they report sensations, such as a touch of the amputated limb, on specific areas of the residual limb [33,34,35,36]. Surgical procedures such as targeted muscle reinnervation (TMR) [37] and rerouting sensory afferent fibers to skin areas during targeted sensory reinnervation (TSR) [17,38] can promote the development of peripheral phantom limb maps, helping to enable bidirectional neural interfaces to both receive information from and transmit information to prostheses [39]. FEELIX is a prosthesis add-on and non-invasive closed-loop somatosensory feedback prototype developed to improve manual dexterity in persons with transradial amputation. Without any surgery required, FEELIX maps pressure signals from the prosthetic hand to vibrotactile feedback on an area the user associates with the sensor locations on the hand.

The restoration of sensory feedback by the FEELIX system would likely enable users to handle objects in their prosthetic hand without relying on assistance from visual feedback or the unaffected hand. To test this hypothesis, we report a series of clinical tests involving a patient with a transradial amputation and concomitant bilateral vision loss, following a traumatic blast injury. We used clinical tests of manual dexterity to investigate the effect of FEELIX, as an add-on to a non-tactile hand prosthesis, on the scores of functional motor tasks. The patient’s traumatic vision loss prevented the use of typical visual compensatory mechanisms, which are commonly used by individuals with loss of peripheral sensory input. Since the patient had no option to compensate or learn through visual feedback, this case demonstrates the unaffected pure impact of restoring somatosensory feedback from a prosthetic hand on improving manual dexterity.

## 2. Detailed Case Description

We describe the case of a 30-year-old Caucasian man with a transradial amputation and vision loss after a traumatic injury. He gave signed authorization to use or disclose health information in compliance with the HIPAA privacy standards and was treated according to Article 37 of the Declaration of Helsinki [40].

### 2.1. Timeline

In April 2021, a firecracker explosion caused second-degree burns, foreign body impactions on the patient’s arms and face, blast trauma on both sides, perforation of both eyeballs and disarticulation of the left wrist (Figure 1A). These injuries led to the transradial amputation of the left forearm (Figure 1B) and complete blindness. At the time of the accident, the patient was engaged to be married, employed full time, and socially well supported.

In June 2021, the patient arrived for a 12-week stay at the rehabilitation center. During the initial stages of rehabilitation, the patient required assistance from a companion to navigate the facility while adjusting to the loss of vision. The in-house orthopedic workshop fitted a myoelectric hand prosthesis with a single grip hand, which enabled him to use a white cane while carrying objects simultaneously. Later, to allow greater functionality of the prosthetic hand, a multi-grip hand with wrist rotation was fitted; however, he still used his unaffected hand to check the handover of objects each time. His participation improved during his time at the rehabilitation facility, and he could move independently without an accompanying person.

In August 2021, a vibrotactile feedback add-on system FEELIX was developed as a prototype add-on to the existing myoelectric exo-prosthetic hand (based on Saphenus Medical Technology’s Suralis^®^ system, Vienna, Austria). The first version, depicted in Figure 2, was a cotton glove with force-sensitive resistors, but it produced frequent false positive signals. This led to the development of a prototype with a silicone glove, finalized in August 2022 and shown in Figure 3.

After a short training phase with the second prototype, the patient underwent a three-week training phase at home. In September 2022, his manual dexterity with the use of FEELIX was assessed. The patient then completed a three-month wash-out period during which he used the prosthetic without FEELIX, and then repeated the manual dexterity assessment in January 2023. The timeline of treatment is presented in Figure 4.

### 2.2. Diagnostic Assessment

To compare the functionality of the myoelectric prosthetic hand with and without the use of FEELIX, as well as to compare to the previous literature, we used the clinically validated Southampton Hand Assessment Procedure (SHAP) and Box and Blocks test, and hypothesized that task completion times with FEELIX would be shorter than without FEELIX (Figure 5). To avoid learning effects favoring FEELIX, we tested the prosthetic functionality with the use of FEELIX first, followed by three months of wash-out (daily prosthetic use without FEELIX) before testing prosthetic functionality again [41]. Videos of the manual dexterity assessments were recorded for review. We compared the SHAP times to completion with and without FEELIX using a paired-sample Wilcoxon signed-rank test (statistics software JASP, version 0.17.1). The SHAP index of hand function and the Box and Blocks test results are presented descriptively, as statistical analysis is not feasible for single numbers without applicable minimal detectable change values for this specific population.

#### Manual Dexterity Tests

The SHAP tests the effectiveness of upper-limb prostheses during manipulating six lightweight and heavy abstract objects and 14 activities of daily living, with each task timed by the participant. By measuring SHAP completion times, it is possible to evaluate overall hand function and categorize it into six prehensile patterns. Scores above 100 indicate outstanding hand function, while scores below 100 suggest a decreased hand function compared to healthy persons [42]. The SHAP provides a comprehensive assessment of hand function by encompassing a spectrum of tasks, from abstract to practical, and evaluating a range of motor skills, from fine to gross. The Box and Blocks test measures gross manual dexterity, which refers to the ability to perform large, coordinated movements involving the use of the arms, hands, and fingers. The test requires participants to transfer as many cube-shaped wooden blocks as possible, one at a time, from one side of a box to the other within one minute [43]. While the Box and Blocks test is typically not sensitive enough to study the quality of haptic feedback [44], we chose the test due to its straightforward task and to obtain a measure for basic grasping and holding of objects for this case with combined unilateral transradial amputation and blindness.

### 2.3. Technical Design

The vibrotactile feedback system FEELIX converts pressure signals from the prosthetic hand (Figure 3A) into haptic feedback through vibration actuators (Figure 3B). FEELIX consists of two wirelessly connected main components: a sensor module unit with a silicone glove and a central module unit that controls three vibration actuators. The sensor module unit converts analog signals from six resistive pressure sensors, with a range of 5 to 200 N (textile pressure sensor prototype; WEARIC Texible Gmbh, Dornbirn, Austria), at the prosthetic hand’s index finger, thumb, and hypothenar, to digital signals. It streams them via energy-efficient Bluetooth communication to the central module unit. The central module unit controls three vibration actuators (3 V DC encapsulated NFP-E0720 7 mm diameter and 20 mm length vibration motors, with a rated speed of 12,000 ± 2500 rpm and a maximum intensity of 4.5 G; Need For Power Electronics Co, Kowloon, Hongkong) to provide haptic feedback (Figure 2 and Figure 3). The central and sensor module units include Nordic^®^ nRF52840 microcontrollers with 32-bit ARM^®^ Cortex™-M4 CPUs. Depending on use, the wirelessly charged 3.7 V LiPo batteries usually last up to one week in the central unit and up to 5 days in the sensor unit. We use proprietary algorithms to process the sensor data and to control the vibration actuators. The actuators were positioned outside the socket at the upper arm as a pure add-on system. We selected actuator locations based on ergonomic aspects, meridian paths originating from traditional Chinese medicine, and suggestions from the participant, optimizing his ability to differentiate the regions. Thus, we placed the thumb actuator on the anterior side of the middle third of the upper arm, the index finger lateral, and the hypothenar actuator dorsal to the thumb placement.

The vibrations synchronize with an approximately 100 ms delay to the pressure sensors. The motor rotation-dependent vibration frequency and intensity are individually adjusted together with the vibration period for each patient via a mobile application and, once triggered, the vibration is either on or off; thus, it does not adapt to changes in pressure. Suppose one of the two assigned pressure sensors per region (thumb, index finger, or lateral hand) detects pressure above an individually preset 10 N to 20 N sensor threshold. In that case, the corresponding motor vibrates, providing haptic feedback to the user. The motor has a maximum vibration period of 750 ms. If both sensors of a region are below the individually preset threshold before the preset maximum time has elapsed, the motor will stop vibrating earlier. Based on the individual preset, the fixed vibration frequency is set to a value between 50 Hz and 70 Hz, corresponding to an intensity value between 40% and 75%, which is usually preferred by users of the Suralis^®^ system.

### 2.4. Results of Manual Dexterity Tests

The overall time to SHAP completion improved by 26% from 9.8 min without FEELIX to 7.3 min with FEELIX, as shown in Table 1 (z = −3.371, *p* < 0.001, effect size = −0.79 with [-inf, −0.591] 95% CI, Figure 6).

The SHAP index of function increased from 30% without FEELIX to 43% with FEELIX, where 100% represents healthy persons (Table 2) [45].

The Box and Blocks results in Table 3 show that two out of three attempts at grasping and transporting cubes failed without FEELIX or unaffected hand assistance. This could be compensated for by checking the gripping of objects with the unaffected hand or with FEELIX.

## 3. Discussion

Providing vibrotactile feedback from the myoelectric multi-grip prosthetic hand to the residual upper arm of a person with transradial amputation and concomitant traumatic vision loss led to improved scores of clinical tests of manual dexterity. The participant’s SHAP overall score of hand function improved from 30% to 43% (healthy individuals score is 100%). Overall functional scores in people with transradial amputation without vision loss range between 24.8% and 54.4%, with an average score of 36.9%, with the use of myoelectric multi-grip hands [46]. FEELIX allowed the participant to shift from a below-average functionality to an above-average functionality score for people with transradial amputation without vision loss, suggesting that the sensory feedback system could improve activities of daily living involving fine hand and finger movements. The effect of the sensory feedback system was especially noticeable when handling lightweight abstract objects (LAO), and when performing activities of daily living such as operating a zipper, which improved by 37%, and rotating a screw 90 degrees (Figure 6), which could not be completed with the prosthetic hand without the use of FEELIX. Previous studies have shown that experienced prosthetic users with transradial amputation without vision loss have a median SHAP completion time of 27.5 s in the SHAP LAO test and completed the zipper and screw tests in 19.2 and 31.2 s, respectively [47]. In the current study, we found the patient increased their LAO task completion time from 75.0 s without FEELIX to 45.4 s with FEELIX. The SHAP LAO completion time is the sum of completion times for six individual tests. In the current study, the patient had longer completion times in individual tasks as a result of their vision loss, and possibly also as a result of having less experience using a prosthetic hand than subjects in previous studies. For example, the patient had difficulty handling small objects like coins and buttons due to a lack of perception, and as a result, had to utilize the unaffected hand to compensate for the absence of visual aid that even people without amputation use for such tasks. Altogether, use of the FEELIX system improved the SHAP overall hand function score compared to the use of a myoelectric multi-grip prosthetic hand without FEELIX in a person with transradial amputation and concomitant vision loss.

When performing the Box and Blocks test of gross manual dexterity, we found that the patient was able to transport six blocks per minute without assistance from the unaffected hand, while using the FEELIX system. In comparison, we found that the patient transported four blocks per minute without either assistance from the unaffected hand or the use of FEELIX, and failed an attempt to grasp and move blocks eight times per minute. Mathiowetz et al. [48] reported an average of 81.3 cube transports per minute in healthy persons 30–34 years old using their left hand. Using an implanted peripheral nerve interface to elicit frequency- and amplitude-modulated sensations of touch and magnitude of contact force, Valle et al. [49] reported an increase in the number of fragile blocks transported per minute from 5.8 without feedback to 9.5 with feedback and an increase in success rate from 45% to 70% in one person with transradial amputation. Similarly, George et al. [44] reported an increase in the number of fragile blocks transported per minute from 5.4 without sensory feedback to 6.6 with sensory feedback and an increase in success rate from 55% to 80% in a subject with transradial amputation using an intraneural force-encoding stimulation. Although comparison of an implanted interface to an external assistive device such as FEELIX may lack validity, the results of the study present valuable information regarding function outcomes when providing sensory feedback to prosthetic users with transradial amputation. The presence of vision loss typically results in enhanced passive tactile acuity compared to sighted individuals, but the combination with transradial amputation likely potentiated the complexity of the basic Box and Blocks test, which is typically not sensitive enough to measure tactile sensitivity in sighted people [9,44]. However, we observed that the success rate for transporting blocks increased from 33% without FEELIX to 100% with FEELIX, suggesting that the use of vibrotactile systems with myoelectric multi-grip hands can provide reliable feedback on grasping and holding objects. Similar to the more complex approaches, our findings demonstrate the potential of non-invasive vibrotactile feedback in restoring basic grasping and holding abilities. However, the results are likely influenced by sight, suggesting sensory feedback systems such as FEELIX would produce more pronounced effects for users with vision loss compared to sighted users.

The participant described the subjective ergonomic benefits of the vibrotactile feedback as follows:
“The tactile feedback is a great help to me. I no longer have to constantly check whether I’m holding something with the prosthesis. In my spare time, I love building nest boxes for birds. FEELIX makes work processes safer and easier. I also get a better feel for my hands and can work with both hands again”, a summary of doctor’s visits (10/2022).

It is important to note that one and a half years after the accident, and after the assessments described in the report, the patient developed a peripheral phantom hand map on the inner forearm of the affected upper limb without surgical intervention, following continued use of FEELIX on his own account at home. The index finger and thumb could be clearly distinguished from each other on the forearm, with areas of about two square centimeters, which thus required actuators to be relocated to these areas. It is vital to consider phantom limb maps when restoring sensation in people with limb amputations. Previous studies have shown that providing proprioceptive and tactile feedback to peripheral phantom hand maps can reduce dependence on visual control [50], improve touch thresholds and discriminative touch [33], and correctly identify touch on forearm sites [12,39]. Further, the related cortical reorganization [33] and the subsequent recovery of the physiological organization and function of the somatosensory and motor cortex network can drastically reduce phantom limb pain in the long term [51,52,53]. Additionally, reducing pain and activating the ascending arousal system can improve agency and embodiment through restored sensation, and promote physical activity and active participation in social life for the user [54,55,56,57].

There were some limitations to the system and study. The functional outcomes of vibrotactile feedback in conjunction with prosthetic use are complex and can depend on factors such as the level of amputation, prosthetic device type, and user preferences [3]. The current case study only measured outcomes from one participant with a transradial amputation, and thus, the findings cannot be extrapolated to other types of upper-limb amputation. Additionally, the prototype could have been more user-friendly and less obtrusive to improve its suitability for daily use at home. Increasing the contact sensing area and miniaturizing the system would improve its suitability for everyday use.

Further research is necessary to understand the effect of sensory feedback systems such as FEELIX on manual dexterity metrics for sighted users. Studying the effects of sensory feedback on other levels of upper arm amputations, lower-limb amputations, or people with neurological disorders such as polyneuropathy would broaden the application of such devices and assist more people in regaining limb function. A large-scale, multi-center study should focus on the prevalence and quality of phantom hand maps and whether sensory feedback systems and reinnervation surgeries correlate with their development and sensory quality. This should investigate if the modulation of vibration amplitude, frequency, and force can encode grip-contact force and object weight. Such research could uncover potential benefits and challenges for optimizing the design and implementation of future non-invasive sensory feedback systems.

## 4. Conclusions

The results of the current case study offer further support that vibratory stimulation of the sense of touch has the potential to positively and systemically affect the central nervous system of people with transradial amputation with or without vision loss [26,32,58,59]. Sensory feedback from the prosthetic fingers and hand improved the manual dexterity of the myoelectric multi-grip prosthetic hand by improving grasping ability and coordination in a person with transradial amputation and concomitant vision loss. Due to the complex hybrid and myoelectric multi-joint control approaches in transhumeral amputations or shoulder disarticulations [60], the usability of FEELIX could be limited in such cases. Additionally, since people with visual loss have enhanced proprioception, haptic judgment, and passive tactile acuity compared to sighted individuals, the effect of providing sensory feedback from prosthetic hands may not be as pronounced in sighted people [9]. However, it is likely that the FEELIX vibrotactile system, when used in conjunction with a myoelectric multi-grip prosthetic hand, would improve manual dexterity, and thus the ability to perform activities of daily living, by increasing peripheral sensitivity through vibratory stimulation in people with transradial amputation with and without vision loss.

## Figures and Tables

**Figure 1 medicina-59-01710-f001:**
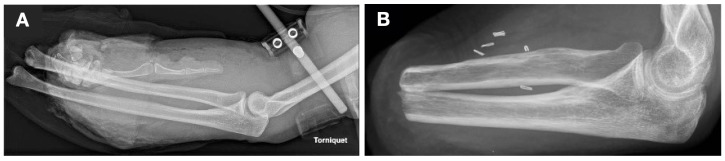
X-rays of the affected left forearm (**A**) shortly after the explosion and injury (3 April 2021) and (**B**) five months after transradial amputation (vessel clips visible on the X-ray from 21 September 2021).

**Figure 2 medicina-59-01710-f002:**
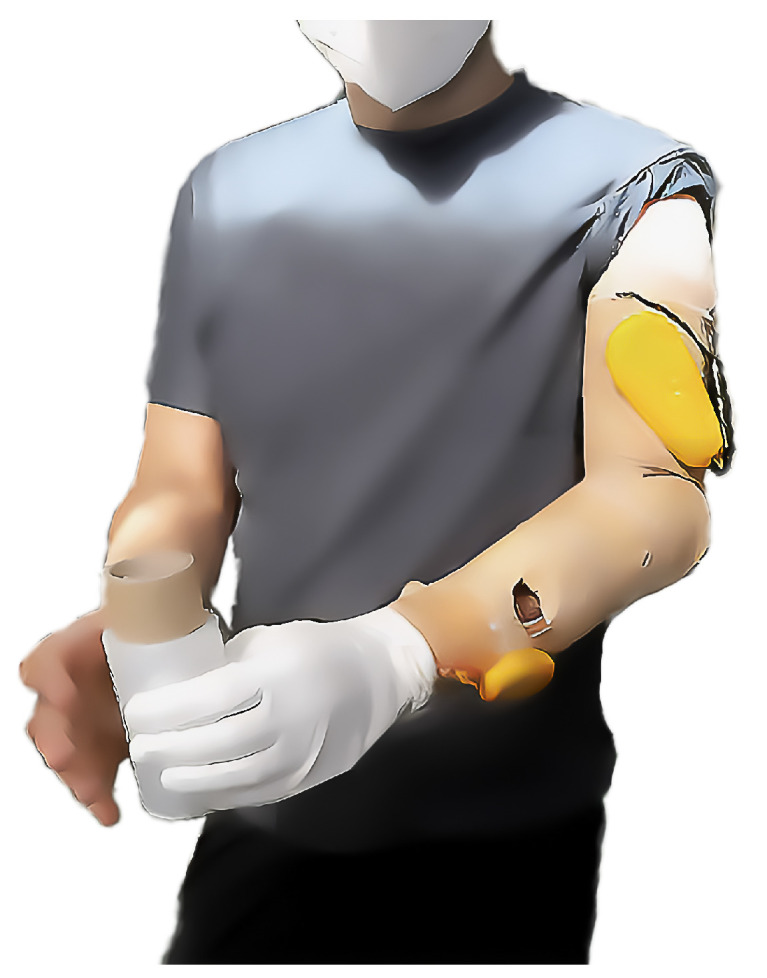
The participant with the first cotton glove-based prototype of the tactile feedback system FEELIX, shown to demonstrate external device components and prosthetic fit.

**Figure 3 medicina-59-01710-f003:**
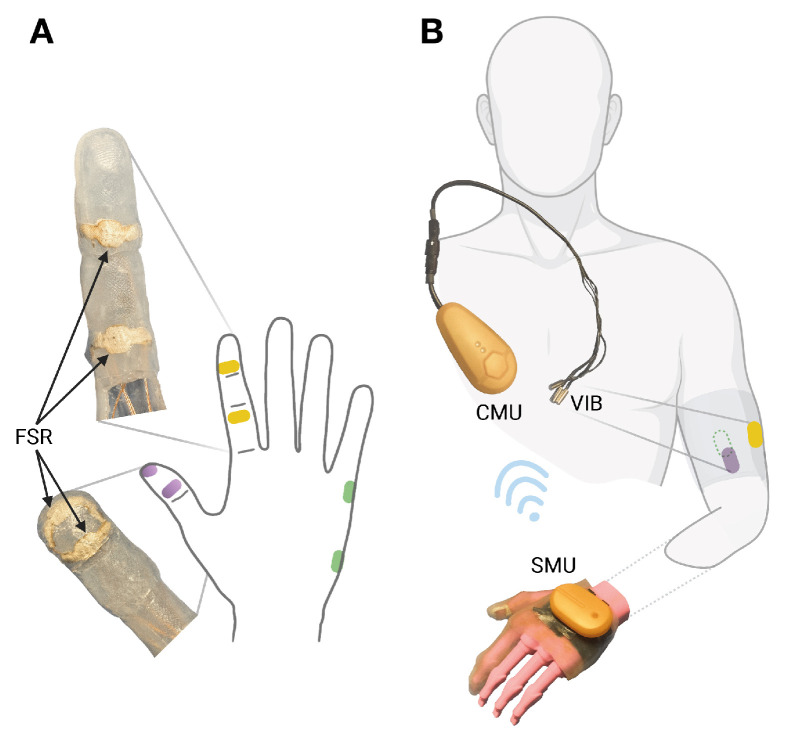
FEELIX as silicon glove. (**A**) Force-sensitive resistors (FSRs) on the prosthetic index finger (yellow), thumb (purple), and hypothenar (green). (**B**) The sensor module unit (SMU) wirelessly transmits force data to the control module unit (CMU), controlling vibration feedback (VIB) on the front (purple), back (green), and lateral (yellow) sides of the residual upper arm.

**Figure 4 medicina-59-01710-f004:**
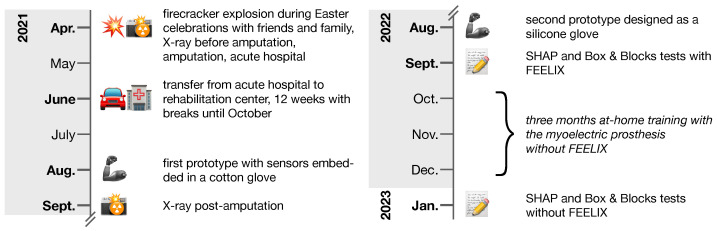
Timeline of key events from accident to Southampton Hand Assessment Procedure (SHAP) and Box and Blocks assessments. During the 12-week rehabilitation center stay, from June 2021 until October 2021, the patient received occupational therapy, physiotherapy, medical training therapy, strength training, hydrotherapy, electrotherapy, scar massage, paraffin packs, laser therapy, psychological care, and social counseling, along with regular consultations with plastic surgery, ophthalmology, and psychiatry specialists. The assessments of manual dexterity were conducted following the completion of in-hospital rehabilitation.

**Figure 5 medicina-59-01710-f005:**
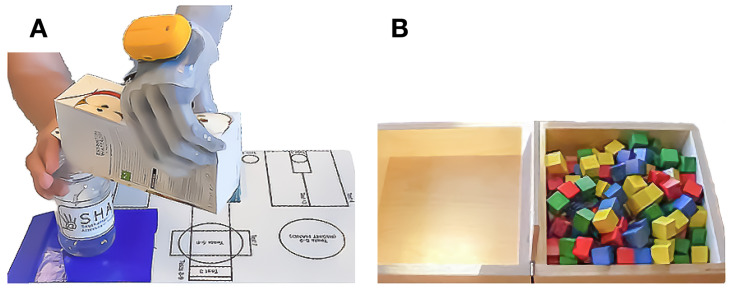
Assessments of manual dexterity. (**A**) The participant successfully managed the Southampton Hand Assessment Procedure activity of daily living ”carton pouring”. (**B**) During the Box and Blocks test, the participant transported cubes from one side to the other, one cube at a time, within a minute.

**Figure 6 medicina-59-01710-f006:**
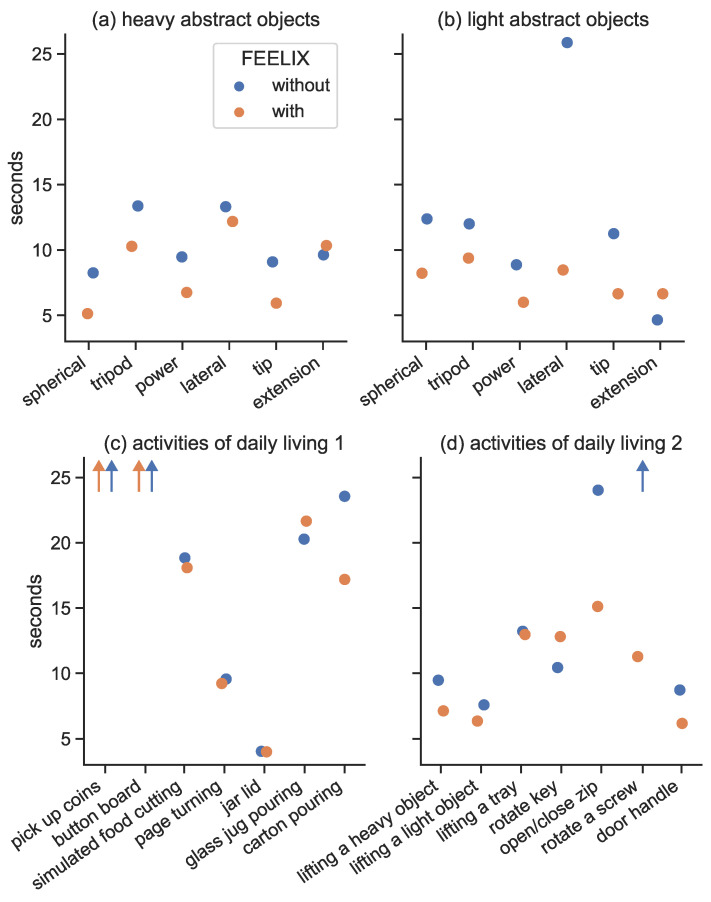
The figure shows the results of the Southampton Hand Assessment Procedure (SHAP) for manipulating heavy abstract objects (**a**) and light abstract objects (**b**), as well as for activities of daily living (**c**,**d**), with and without the use of FEELIX. The orange dots represent the results with FEELIX, while the blue dots represent the results without FEELIX. The improvements in unimanual tasks were more noticeable because, in the absence of FEELIX, feedback was only available through weight-dependent bending forces at the proximal end of the prosthesis. The x-axis displays the different SHAP tasks, while the y-axis shows their time to completion.

**Table 1 medicina-59-01710-t001:** Comparison of total time for each category and overall completion time of the Southampton Hand Assessment Procedure with and without FEELIX.

Timed Tasks	With (s)	Without (s)
Heavy abstract objects	50.6	63.1
Light abstract objects	45.4	75.0
Activities of daily living 1	270.2	276.3
Activities of daily living 2	71.8	173.5
Overall total time	437.9	587.9

**Table 2 medicina-59-01710-t002:** The normalized completion times of the six prehensile patterns in the Southampton Hand Assessment Procedure are merged to calculate the overall hand index of function with and without FEELIX.

Functionality Profile	With (% Healthy Controls)	Without (% Healthy Controls)
Spherical	65	43
Power	57	34
Tip	18	10
Tripod	23	12
Lateral	34	24
Extension	45	47
Index of function	43	30

**Table 3 medicina-59-01710-t003:** Results of the Box and Blocks test for one attempt with and without using the FEELIX tactile feedback system.

	With		Without	
	With Unaffected Hand Assist	Without Unaffected Hand Assist	With Unaffected Hand Assist	Without Unaffected Hand Assist
Cubes per minute	11	6	11	4
Failed attempts per minute	0	0	0	8
Comments	No checking with the unaffected hand necessary	Exact grasping is evident	Checking with the unaffected hand necessary	No exact grasping

## Data Availability

All data are contained within the article.

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
