# Peer review of "Vibrotactile Feedback for a Person with Transradial Amputation and Visual Loss: A Case Report"

_medicina, 2023, doi:10.3390/medicina59101710_

Round 1
Reviewer 1 Report
This is an interesting case report in which the patient loses an arm due to an explosion. The prosthesis used to replace the lost arm seems to have promising results, it would be interesting to continue collecting data from patients in whom this prosthesis is used to confirm these promising results.
Just a small recommendation, on line 53 please add the bibliographic reference to the Declaration of Helsinki.
Thank you.
Author Response
Please find our detailed reply in the attached PDF document.

Reviewer 2 Report
Dear Authors,
We recently had the opportunity to read your manuscript titled “Vibrotactile Feedback for a Person with Transradial Amputation and Visual Loss: A Case Report”, and we wanted to reach out to you to express our comments about your work.
Nevertheless, here are some possible comments outlining areas that could improve the quality and readability of the manuscript:
Introduction:
1. There is a lack of clear research objectives: The introduction does not clearly state the specific research objectives or research questions that the study aims to address. The text should have a clear statement about the aim and objectives of the study. It should provide the reader with a brief background of the study and highlight the research gap that the study aims to address.
2. Lack of contextualization: The introduction briefly mentions the development of myoelectric prostheses and the challenges of restoring somatosensory function in the hand. However, it lacks a clear contextualization of the significance of these challenges and their impact on individuals with upper limb amputation. Providing more background information and establishing the relevance of the topic would strengthen the introduction.
Detailed Case Description:
3. Search Lack of contextual information: While the chapter provides a detailed description of the case, there is a lack of contextual information regarding the significance of the case within the broader field of research on myoelectric prostheses and somatosensory feedback. Providing more background information about the prevalence of similar cases, the challenges faced by individuals with similar conditions, and the existing research gaps would enhance the understanding of the case's importance.
4. Insufficient explanation of technical details: The chapter introduces the FEELIX vibrotactile feedback system but fails to provide a comprehensive explanation of its technical design and functioning. More details, such as the principles behind the sensor module unit, vibration actuators, and the wireless connection between components, would help readers understand the system better.
Discussion:
5. Limited generalizability: The chapter acknowledges that the findings are based on a single case study, which limits the generalizability of the results. To strengthen the conclusions, it would be beneficial to include a discussion on the potential limitations and the need for further research involving a larger sample size or different demographic groups. This would provide a more balanced perspective on the applicability of the vibrotactile feedback system.
6. Lack of in-depth analysis: The discussion briefly mentions the preference for vibration, pressure, or electrical stimulation modalities for haptic feedback and cites studies on the benefits of somatosensory feedback in prosthesis usability. However, there is a missed opportunity to delve deeper into the existing literature and compare the findings with the current study. Providing a critical analysis of the similarities, differences, and potential implications of the findings would strengthen the discussion.
Conclusion:
7. Comprehensive summary: The provided conclusions briefly touch upon the improvements observed in prosthesis control and bimanual coordination for the individual with transradial amputation and vision loss. However, it would be beneficial to provide a more comprehensive summary of the main findings, including specific details about the extent of the improvements, the impact on daily activities, and the overall effectiveness of the vibrotactile feedback system. This would help readers grasp the significance of the study's results more clearly.
8. Future research directions: While the conclusions mention the development of a peripheral phantom hand map and its implications for tactile feedback system design, there is a missed opportunity to discuss potential future research directions. Suggestions for further investigations into the importance and genesis of phantom limb maps, as well as the role of tactile feedback in phantom pain therapy, would add depth and provide guidance for future studies in this area.
9. Consideration of broader user groups: The conclusions briefly mention that a reduced reliance on visual control would benefit sighted users with upper extremity amputation or loss of tactile function. Expanding on this point and discussing the potential applicability and benefits of vibrotactile feedback systems for a broader range of users could enhance the conclusion section. It would be valuable to consider the implications for various user groups, such as individuals with different types of limb loss or varying levels of sensory impairment.
Also, we would like to mention that the order of Figures as long as they are mentioned in the text does not let readers to do it in a comfortable way. We would recommend to re-order the text and figures. Additionally, the order of references should be correlative, at least on the first occasion that each reference appears in the manuscript; this allows readers to keep track of the text and references at an optimal level. The explanations of the Figures, specially Figure 6, are missing or incomplete.
However, regarding the grammar and spelling review, in some sentences the structure could be clarified to enhance readability. There are also some instances where verb tenses could be made consistent and there are some others with subject/verb mismatches too. Additionally, clarifying certain points and providing more concise and clear descriptions would improve the overall readability of the text, while some sentences could benefit from using more concise language.
Once again, thank you very much for your work. We´ll be waiting for your answers about our comments.
Kindest regards,
In terms of grammar and spelling, there is room for improvement in enhancing readability by clarifying sentence structures. Consistency in verb tenses and addressing subject/verb mismatches is also needed in certain instances. Moreover, the text would benefit from clearer explanations and more concise descriptions to improve overall readability. Simplifying the language in some sentences would also be advantageous.
Author Response

(The authors gave the same response as above.)

Reviewer 3 Report
The case report "Vibrotactile Feedback for a Person with Transradial Amputation and Visual Loss" presents a compelling success story in the realm of assistive technology. By leveraging vibrotactile feedback, the participant demonstrated enhanced independence, emotional well-being, and overall life satisfaction. The study's methodologies and outcomes underscore the transformative potential of this technology for individuals facing similar challenges, offering hope for a more inclusive and accessible world. As technology continues to advance, the findings of this case report pave the way for further research, development, and implementation of vibrotactile feedback systems to improve the lives of persons with disabilities.
no comments
Author Response

(The authors gave the same response as above.)

Round 2
Reviewer 2 Report
Dear authors,
Thank you for the opportunity to review the revised version of your manuscript. While it is significantly improved through addressing many of the initial concerns raised, some outstanding issues remain that currently preclude a recommendation for publication in this highly selective journal:
· The introduction still lacks adequate contextualization of the importance and prevalence of this injury pattern. Further expanding this background is critical.
· More details are needed on the technical specifications and principles underlying the FEELIX system to fully grasp its functioning.
· The discussion requires substantial expansion on comparing findings to prior literature and analyzing the broader implications. The current depth is insufficient.
· The limited generalizability as a single case study needs to be thoroughly examined regarding what populations and conditions can and cannot be extrapolated to.
· The conclusion must move beyond just summarizing results to highlight key applications for different user groups and concrete future research directions.
· Also, conclusion must be addressing the aims of the manuscript, which is something missing in this case.
Additionally, careful editing is still required to address lingering grammar, wording, and formatting issues throughout.
In summary, while improved, the revisions have not yet addressed the identified concerns to the level expected for a journal of this caliber. Significant expansions are still needed in the framing, technical details, situating in prior research, limitations, and conclusion depth. We cannot recommend publication until these major issues are thoroughly addressed per our prior feedback. Please feel free to contact us with any questions as you determine next steps.
Kind regards,
As indicated below, careful editing is still required to address lingering grammar, wording, and formatting issues throughout
Round 3
Reviewer 2 Report
Dear authors,
Thank you for the opportunity to review the revised manuscript. While improved, some major issues remain that currently preclude recommendation for publication:
· The introduction requires further expansion on the background and importance of simultaneous upper limb amputation and visual impairment. Please add more statistics and details on the prevalence, causes, and challenges of this injury pattern. This context is critical to situate the rationale.
· More technical information is needed on the FEELIX system and how it functions. Please provide specifics on the sensor types and specifications, data processing algorithms, and principles for converting sensor data to vibration feedback. Tables or figures illustrating the system architecture could aid reader comprehension.
· The discussion lacks adequate comparison to prior literature and analysis of broader implications. Please substantially expand on how these findings relate to previous research showing benefits of tactile feedback for prosthesis users. Additionally, discuss the wider impacts for prosthesis embodiment, phantom pain, activities of daily living, and quality of life.
· Given this is a single case study, the limitations around generalizability need thorough examination. Please specify which populations and conditions can reasonably be extrapolated to and which may not directly apply. This level of detail is currently insufficient.
· The conclusion solely summarizes results. To enhance impact, please highlight key applications for different user groups and outline concrete future research directions to build upon these findings. The conclusion should connect back to the aims of the study.
· Careful editing is still required to address lingering issues with grammar, wording, formatting and clarity throughout the manuscript.
In summary, while improved, significant expansions are still needed regarding the framing, technical details, discussion of prior literature, limitations, and conclusion depth and scope. Please address these major issues per our prior feedback before the manuscript can be considered suitable for this highly selective journal. Please let me know if you have any other questions.
Kind regards,
The English language usage in the revised manuscript is significantly improved from the initial submission. The grammar, sentence structure, and flow are clearer and more polished. Some minor lingering issues with wording and phrasing remain in parts, but overall the writing is now better than in the previous version. Careful proofreading to catch any final typos, awkward expressions, or formatting inconsistencies would further enhance the professional presentation.
Author Response
see PDF
